# Preoperative Clinical Phenotyping for Individualised Rehabilitation in End-Stage Knee Osteoarthritis

**DOI:** 10.3390/jfmk10030360

**Published:** 2025-09-19

**Authors:** Marisa Coetzee, Amanda Marie Clifford, Diribsa Tsegaya Bedada, Oloff Bergh, Quinette Abegail Louw

**Affiliations:** 1Division of Physiotherapy, Department of Health and Rehabilitation Sciences, Stellenbosch University, Cape Town 7599, South Africa; 2School of Allied Health, Health Research Institute, Ageing Research Centre, University of Limerick, V94 T9PX Limerick, Ireland; 3Department of Statistics and Actuarial Science, University of Waterloo, Waterloo, ON N2L 3G1, Canada; 4Neuromechanics Unit, Central Analytical Facility, Stellenbosch University, Cape Town 7599, South Africa

**Keywords:** clinical phenotype, factor analysis, chronic pain, total knee replacement, osteoarthritis, movement analysis

## Abstract

**Background:** Osteoarthritis (OA) of the knee is a highly prevalent and heterogeneous condition. Identifying distinct clinical phenotypes within end-stage knee OA populations may inform tailored preoperative management strategies for individuals awaiting total knee replacement (TKR) surgery. **Methods:** This cross-sectional study employed exploratory factor analysis to identify clinical presentation patterns among patients with knee OA awaiting TKR in South Africa, using modifiable variables including demographic data, physical examination findings, patient-reported outcomes, and functional measures. **Results:** Three distinct clinical phenotypes emerged: (1) gait and weight—characterised by poor gait mechanics, obesity, and low self-efficacy; (2) central pain—encompassing central sensitisation, depression, and reduced functional performance; and (3) functional factors—reflecting muscular weakness and functional limitations. **Conclusions:** This study highlights the heterogeneity in clinical presentations among patients with end-stage knee OA awaiting TKR in South Africa. The identified phenotypes suggest a need for tailored, multidisciplinary preoperative interventions incorporating weight management, pain management, psychological support, targeted exercise programs, and behavioural change strategies to optimise post-surgical outcomes and enhance overall care.

## 1. Introduction

Osteoarthritis (OA), particularly knee OA, is a highly prevalent global musculoskeletal condition, characterised by substantial heterogeneity in its clinical manifestations, disease trajectories, and underlying pathophysiological mechanisms [1]. Emerging evidence highlights significant inter-individual variability in symptom severity, disease progression, and treatment response among patients with knee OA [2]. This heterogeneity is influenced by factors such as age, sex, genetics, biomechanics, comorbidities, and lifestyle, which influence the individual response to medical management and reduce the efficacy of standardised therapeutic interventions [1].

In recent years, the field of phenotype research has gained prominence, aiming to explain the heterogeneity of knee OA through advanced imaging modalities, biomarker analysis, and data-driven analytics [3]. By identifying distinct subgroups or phenotypes within the broader knee OA population, characterised by unique clinical, radiographic, and molecular signatures, phenotype research holds promise in clarifying underlying pathogenic mechanisms, predicting disease progression, and tailoring interventions to specific phenotypic profiles [2,4,5,6,7,8]. Although phenotypic classification based on molecular characteristics may provide detailed insight into underlying biological pathways, cellular taxonomy, and biochemical markers [9], it requires laboratory analysis, which may be impractical in low-resource clinical settings. In contrast, phenotyping based on clinical characteristics may be less detailed [10], but more practical in clinical settings.

Clinical phenotyping involves grouping patients based on observable and modifiable clinical characteristics, such as posture, gait, physical activity levels, obesity, muscle strength, balance, and pain behaviour [5,11,12,13]. Commonly included variables span domains such as age, sex, body mass index (BMI), range of motion, muscle strength, gait analysis, pain severity, psychological factors (e.g., depression, anxiety), self-efficacy, and functional performance (e.g., sit-to-stand tests, physical activity levels) [11,12]. Although studies looking at the effects of phenotype-specific interventions are still in their infancy, promising results for pain and functional response to exercise interventions have identified baseline factors related to trajectories with different treatment responses [4,7,14]. By stratification of distinct clinical phenotypes, targeted management strategies can be developed to address the unique needs of each subgroup.

Current management strategies for OA include pharmacological (e.g., NSAIDs, corticosteroids), non-pharmacological (e.g., exercise, patient education), and end-stage surgical approaches. Disease-modifying OA drugs (DMOADs) targeting specific molecular pathways are also being explored, with the intention to eventually be able to slow down, halt joint degeneration, or even reverse the effects of OA [15]. Management is typically focused on relieving pain and improving joint function, as these are reported to be the most disabling symptoms [15]. In OA patients, pain severity does not always correlate with radiographic changes [16]. Emerging evidence suggests that neuroinflammation, central sensitisation, and maladaptive pain processing contribute to chronic pain in patients with OA. Patients with chronic pain are at risk of developing central sensitisation and nociplastic pain, which can lead to poor postoperative outcomes [15,17]. In addition, pain and reduced function can lead to physical deconditioning, obesity, and the development of cardiovascular and metabolic conditions, which can negatively affect the pain experience and increase the risk for poor postoperative outcomes [18].

In South Africa, the public healthcare sector is burdened by an increase in patients with end-stage knee OA awaiting TKR surgeries [19]. Average waiting times of 3.5 to 5 years have been reported, and a majority of patients were not referred for rehabilitation interventions during this time [20]. In other contexts, prolonged surgical waiting times have been shown to exacerbate functional decline, pain, and psychosocial distress, potentially compromising postoperative outcomes [21,22]. Identifying the modifiable clinical presentation patterns in patients with end-stage OA can facilitate targeted rehabilitation interventions to ensure that patients have better pre- and postoperative outcomes and that resources are allocated optimally, especially in low- and middle-income countries like South Africa [10]. However, few studies on patient clinical presentation profiles have been conducted in the South African context. Therefore, the aim of this study was to identify distinct clinical phenotypes in the South African population to inform tailored preoperative rehabilitation strategies.

## 2. Materials and Methods

### 2.1. Study Design

A descriptive cross-sectional study design was used. All the individuals on the waiting list at Tygerberg Hospital were invited to take part in a previous study [19] and those who gave consent to take part in further studies were included in the sample pool for this study.

### 2.2. Study Participants

Conducting sample size calculation for factor analysis is complex due to the lack of a priori hypothesis (not knowing if, what, or how many factors there will be) [23]. Rule-of-thumb estimates have been reported in the literature for use within factor analysis and informed the sample size estimation in this study. According to Hair [23], the minimum recommended ratio for sample size in factor analysis is 5:1 (at least 5 individuals per variable), which totalled 65 individuals for the 13 variables we planned to include. A stratified random sampling technique was used [24,25], and the inclusion criteria were being waitlisted for TKR and the knee OA being the primary site of concern (other joints having only mild to moderate OA). Individuals with all levels of mobility (including wheelchair bound patients) were included in this study, and the only exclusion criterion was the inability to give consent. A detailed methodology for this study is presented in Appendix A.

### 2.3. Study Procedures

All attempts were made to minimise the direct financial cost to the participants with regard to time taken from work or their daily activities. Individuals were offered transportation to and from the assessment facility or reimbursed for any travelling costs. In addition, individuals received remuneration for their time, an education session, and a leaflet with information on symptom management. Participants were also offered the option to bring their carer/spouse with them for support. The primary researcher (an experienced physiotherapist and clinical educator) welcomed the participants, provided an overview of the study (including the latest results from the project), and explained the procedure of the session. Followed by a detailed explanation of the self-reported questionnaires, which were completed by the individuals at their own pace during their time at the facility. During the completion of the self-reported questionnaire, assistance was provided by the primary researcher or a research assistant and included reading the questions when the participant forgot their reading glasses, as well as explaining any concepts or scales on the questionnaire. The primary researcher conducted all the objective assessments. All efforts were made to always ensure the safety of participants with ample chairs for resting and the research assistants standing in close proximity at all times.

### 2.4. Outcome Measures

Clinical, functional, biomechanical, and self-reported outcome measures were chosen based on systematic reviews and feasibility studies on structural/functional disease progression and patient subgroups in OA [5,26,27]. In addition, the literature on supported self-management for OA and other chronic conditions was consulted for measures related to important components of self-management such as coping behaviour and self-efficacy [28,29]. Clinical measures included height and weight for body mass index (BMI) calculation; knee range of motion (ROM) flexion and extension, which were measured using a standard goniometer; and muscle strength of knee extensors, which was measured using a Lafayette Handheld Dynamometer following a standardised protocol (HHD). The standardised protocol included three measures of each ROM and muscle strength per measurement, and the average of the three measurements was used to ensure inter-tester reliability. Our functional measure was the 30 s chair stand test (30 s CST), and the biomechanical measure used was the gait deviation index (GDI) [30,31]. Seven self-reported outcome measures were used. They included the Central Sensitisation Inventory (CSI) [32], the 10-item Center for Epidemiologic Studies Depression Scale (Ces-D 10) [33], the patient specific functional scale (PSFS) [34], the Oxford knee score (OKS) [35], the Brief Cope questionnaire [36], the arthritis self-efficacy 8-item scale (ASES-8) [37], and the Global Physical Activity Questionnaire [38]. Details on the outcome measures are presented in Appendix A.

### 2.5. Movement Analysis

All patient-reported outcome measures were printed on paper, and individuals were assisted by research assistants in completing these in their language of choice. A VICON Optical Motion Capture (OMC) system (version 2.11; Vicon Motion Systems Ltd., Oxford, UK), specifically the MX T-series with the Plug-in-Gait (PiG) model, was used for data collection. This advanced system utilises multiple high-resolution cameras to precisely capture and reconstruct marker trajectories, providing detailed information on body segment positions and orientations relative to a global coordinate system. Renowned for its accuracy and reliability, the VICON system, along with the Conventional Gait Model in PiG, is extensively validated and commonly used in clinical research [39]. The setup included eight VICON T-20 infrared cameras (Vicon Motion Systems Ltd., Oxford, UK) paired with VICON Nexus 2.9.2 software, capturing data at 200 Hz. We used thirty-six 14 mm diameter passive retro-reflective markers that were placed on key anatomical landmarks, with additional markers placed on the iliac crest of the pelvis to enhance tracking. All biomechanical outcomes were calculated using the modified lower body PiG model in the Nexus 2.9.2 software. Additionally, gait events were captured using three time-synchronised, floor-embedded force plates (one FP6090-15 and two FP6040-15 models from Bertec Corporation, Columbus, OH, USA).

### 2.6. Data Analysis

Both discrete and continuous numerical variables were collected, alongside nominal, binary, and ordinal categorical variables. Frequency distribution (percentages and counts) and contingency tables were used for reporting on the categorical variables from predetermined answer options, and central tendency was used for reporting numerical variables. Muscle strength percentages of the norm were calculated using normative data per age group. Stata software (StataCorp. 2023. Stata Statistical Software: Release 17. College Station, TX, USA: StataCorp LLC) was used for data management and analysis. Selected variables were subjected to exploratory factor analysis (EFA) for the identification and description of clinical presentation patterns in the population of individuals [23,40]. Missing data were subjected to full information maximum likelihood (only required for GDI, sit-to-stand, and BMI data where individuals were wheelchair bound).

### 2.7. Factor Extraction

As the first step in our EFA, principal component analysis (PCA) has been chosen for factor extraction as it is a frequently recommended method for pattern recognition amongst health-related variables, and it retains the total (both common and unique) variance between the original variables for analysis [23,40]. In addition, an orthogonal factor rotation (promax) was used to reduce ambiguity and improve the interpretation of our data [23]. Each variable had calculated weightings (factor loadings) which showed the relative contribution of each variable to the factor (clinical presentation pattern). Variables with a contribution of 0.3 or more were considered relevant to the factor and were retained [40] for descriptive purposes of the clinical presentation pattern. Variables with a weighting of less than 0.3 were removed from the model. For the factor to be considered a clinical pattern (and based on the proposed sample size), it should have had at least 3 variables contributing 0.3 or more to the factor, and the highest contributing variable should be 0.55 or more for clinical significance [23,40]. The Kaiser–Meyer–Olkin Measure of Sampling Adequacy was 0.5148, which is acceptable for factor analysis, and the data suitability for factor analysis was confirmed with the Bartlett’s test score of <0.0001. In addition, all the factors were subjected to the Kaiser criterion and Joliff’s criterion, which states that the factor must have an eigenvalue of >1 and a mean factor loading of >0.7, respectively, to determine if they are eligible to be retained [40].

### 2.8. Factor Interpretation and Naming

Certain variables are present in more than one factor and were retained in each factor for descriptive purposes, but the factor to which the variable contributes to most was considered more significant for that variable. Variables contributing the most to the factor and are modifiable with rehabilitation were considered a key driver for that factor, and therefore were considered the key focus for intervention within the clinical pattern [23]. The factor compositions were data-driven and based on the loading patterns identified by EFA and were not predetermined. The naming of each factor reflects the variables with the highest loadings and the underlying clinical construct they were judged to represent.

## 3. Results

### 3.1. Demographic and Outcome Measures

A total of 72 individuals participated in the study. Table 1 provides an overview of the results. The majority of participants were female (86.1%) with a mean age of 67.5 years. The mean BMI was 35.8, which is classified as obese, with a standard deviation (SD) of 6.9, placing some individuals in the normal or severely obese categories. The mean gait deviation index was 63 out of 100, showing that, on average, these patients scored below the normal gait parameters, presenting with moderate to severely affected gait patterns. Mean active knee range of motion was 85 (range 73–94), with many individuals achieving less than the safe functional range of 80 degrees. Quadriceps muscle strength was, on average, 35% of the norm for their age and sex. The 30 s chair stand test identified that some patients were unable to complete one repetition during the test, with mean values of 4.6 (SD 3.1), which is less than a third of normal values for healthy individuals.

Most individuals reported an approach coping style as opposed to avoidance, using religion as a means of coping with their pain. Patients also reported a moderate amount of confidence in managing symptoms (using the ASES), with a mean self-efficacy score of 49.5 out of 80. The CSI showed that, on average, patients had moderate levels of central sensitisation, but 40% of patients had severe/extreme central sensitisation. Most patients reported minimal signs of depression; however, 26% of patients had scores that indicated being at risk for clinical depression.

### 3.2. Factor Analysis

The factor analysis produced three distinct clinical presentation patterns (Table 2). The first factor variables included being female, having poor biomechanics of the knee, being obese, having a limited range of motion, and having poor self-efficacy. This variable was named “gait and weight” due to the highest contributing variables being those of obesity and gait deviation. The second factor variables included being male, having a limited range of motion, poor sit-to-stand capability, having central sensitisation, reporting an approach coping style, being depressed, and having severe pain. This variable was named “central pain” due to the highest contributing variables being those of central sensitisation, coping, and depression. The third factor variables include having poor muscle strength, limited physical function, and being sedentary. This categorisation was the “functional” factor due to the presence of only function-related variables.

## 4. Discussion

This study aimed to identify and describe distinct clinical presentation patterns among patients awaiting TKR in South Africa, using modifiable variables that could be targeted with specific management strategies. Our analysis revealed three distinct factors: (1) gait and weight, (2) central pain, and (3) functional. Each of these factors comprises variables that resemble clinical patterns observed in the existing literature.

The first factor, gait and weight, aligns with previously described biomechanical and metabolic phenotypes [12,41,42]. The gait variable, comprising the nine frontal and sagittal knee moments and angles that were calculated in the GDI, provides a numerical value of the amount of deviation from the norm in the gait. This is potentially due to the loss in functional joint range that was seen in this factor, along with the changes in joint structure, which causes deformity and changes the biomechanics of the knee joint as well as the joints above and below [43]. Patients experiencing the most deviation in their gait pattern were likely to be in this group and present with increased weight. Obesity, a significant challenge in South Africa, disproportionately affects vulnerable populations and contributes to altered joint mechanics, increased articular surface loading, and accelerated joint degeneration [42,44]. Addressing obesity requires a comprehensive approach, as outlined in South Africa’s Strategy for the Prevention and Management of Obesity (2023–2028), including equitable access to healthy foods, physical activity opportunities, and a capacitated healthcare system [45]. Having poor self-efficacy was another variable identified in the gait and weight category, meaning patients felt they did not have enough confidence to manage their OA-related symptoms. Notably, research has linked obesity with poor self-efficacy, and enhancing self-efficacy through empowerment and education is critical to sustainable weight management [46]. Hence, comprehensive weight management programs incorporating dietary modifications, physical activity promotion, and self-efficacy enhancement strategies could be beneficial for patients who present with this clinical factor [47].

The central pain factor encompassed variables such as reduced functional performance (e.g., sit-to-stand), central sensitisation, and depression. Self-reported symptoms of central sensitisation, characterised by heightened pain sensitivity due to altered central nervous system processing, were prevalent in a significant proportion (40%) of our sample, higher than typically reported in end-stage knee OA populations (30%) [48,49,50]. Central sensitisation is a result of maladaptive pain processing mechanisms, which cause structural changes in the brain. Coupled with depression and sleep deprivation, it has been argued that the risk for falls increases due to psychomotor retardation, deconditioning, and cognitive impairments [51,52,53]. This subgroup may be more vulnerable to increasing disability and reduction in quality of life while awaiting surgery, as chronic pain and psychological distress can perpetuate a cycle of inactivity, deconditioning, and functional decline [54]. Addressing psychosocial factors and optimising physical function is crucial for successful surgical outcomes in this subgroup, necessitating a staged approach integrating exercise, education, functional activity, and psychological support. Therefore, individuals in this group may benefit from multidisciplinary interventions addressing pain management, psychological support, graded functional retraining, and strategies to break the cycle of inactivity and deconditioning [55,56,57].

The functional factor, characterised by functional limitations, aligns with typical phenotypes identified in previous research that are extra-articular in nature and not driven by molecular mechanisms, but rather as a result of ligament instability, sarcopenia, and non-use [2,3,58]. Joint pathology and degeneration contribute to muscular weakness, particularly in weight-bearing joints, leading to challenges in activities like stair climbing, kneeling, sit-to-stand transitions, and standing up from the floor [59]. Especially in older adults who do not have the upper limb strength to assist them, struggling with transitional functions may lead to falls. The inability to stand up from the floor when they are alone has serious consequences, such as dehydration and hypothermia. Prolonged waiting times may exacerbate muscle atrophy and functional decline due to limited physical activity [60]. For patients within this functional phenotype, preoperative strengthening exercises and behavioural strategies to optimise levels of physical activity and reduce sedentary behaviour may be relatively easy to implement, likely improving postoperative outcomes [61].

Our findings support the need for clinical phenotyping in end-stage knee OA populations, as we identified distinct patterns that may benefit from tailored rehabilitation management approaches. In the South African context, where access to primary care is limited and waiting times for TKR are long [19], identifying clinical phenotypes can inform the development of targeted preoperative interventions. By tailoring rehabilitation strategies to the specific clinical presentations identified through phenotyping, we can potentially optimise preoperative management, improve postoperative outcomes, and enhance overall care for end-stage knee OA patients awaiting TKR in South Africa. Furthermore, early identification of phenotypic profiles could guide timely interventions, potentially delaying disease progression and reducing the need for surgical intervention in some cases.

A Strengthening the Reporting of Observational Studies in Epidemiology (STROBE) statement, which provides guidelines for reporting observational studies, was completed for this study (Appendix A). The limitations of our study include its sole focus on clinical, modifiable variables and excluding genetic, biomolecular, or advanced imaging data. Our study employed a cross-sectional design, capturing a snapshot of the clinical presentations at a single time point. Considering that OA is a progressive condition, longitudinal studies are warranted to evaluate the stability and transition of these phenotypic profiles over time, as well as their potential impact on long-term outcomes following TKR. The study population was limited to patients awaiting TKR in a large hospital in South Africa, potentially introducing selection bias and limiting the generalisability of our findings to broader knee OA populations or different geographic contexts. While we included a comprehensive set of clinical variables, there may be additional relevant factors that could further delineate distinct phenotypes or subgroups within our identified clusters. Future research should consider incorporating a broader range of variables, such as physical activity patterns, dietary habits, or social determinants of health, to provide a more holistic understanding of clinical heterogeneity. In addition, we used no quantitative sensory testing for central sensitisation; therefore, we can only comment on what patients have reported and cannot support the reported scores with sensory testing results. Despite these limitations, our study contributes to the growing body of evidence supporting the value of clinical phenotyping in end-stage knee OA management. In addition, there is potential for some of the results to be generalisable to other LMIC settings when interpreted alongside our previously published work (patient profile description). Future research addressing these limitations, as well as exploring the implementation and outcomes of phenotype-specific interventions, could further advance our understanding and improve the care of individuals living with this debilitating condition.

## 5. Conclusions

In conclusion, this study provides valuable insights into the clinical presentation patterns of patients awaiting TKR. Three distinct factors emerged: gait and weight, central pain, and functional factors, each comprising variables reflective of clinical patterns observed in the literature. Notably, a significant proportion of the population awaiting TKR presents with central sensitisation, depression, and functional impairments in knee range and strength. The gait and weight factor emerged as the strongest, highlighting the impact of obesity on TKR candidacy and outcomes. These findings suggest important considerations for clinical practice. While the data are cross-sectional and do not imply causation, they underscore the need for a multifaceted approach to management that addresses both physical and psychological aspects in this patient population.

## Figures and Tables

**Table 1 jfmk-10-00360-t001:** Summary of the variables assessed.

Variable	n = 72	Missing Data (n)
Sex (female)	62 (86.1%)	
Age (years)	67.5 (SD 7.3)	
Body mass index (kg/m^2^)	35.8 (SD 6.9)	23
Gait deviation index (0–100)	63 (SD 8.7)	23
Range of motion (0–135) degrees (median; IQR)	85 (IQR 73–94)	
Muscle strength (median % of norm; IQR) *	35.1 (26.59–49.22)	
Sit to stand (number of times)	4.6 (SD 3.1)	23
Arthritis self-efficacy score (0–100)	49.5 (SD 17.8)	
Central Sensitisation Inventory (0–80) ^#^	41 (SD 18.2)	
Centre of Epidemiology Depression Score (0–60) ^#^	11.4 (SD 6.2)	
Patient-specific functional score (0–30)	18.3 (IQR 3.33–33.33)	
Oxford knee score (0–48)	28.4 (SD 10.2)	
Physical activity mean (level of activity) **^#^	2.3 (SD 0.8)	

* % of norm as per age and gender. ** Level of physical activity was categorised as 1 (high); 2 (medium/moderate), and 3 (inactive). SD: standard deviation; IQR: interquartile range. ^#^ Higher scores relate to worse outcomes.

**Table 2 jfmk-10-00360-t002:** Factor analysis results with named variables and eigenvalues.

Variable	Gait and Weight	Central Pain	Functional
Gait deviation index	0.7100		
Body mass index	−0.5977		
Range of motion	0.5177		
Muscle strength	0.3695		
Sit to stand			−0.3436
Self-efficacy	0.4980	−0.4891	
Cope_approach			0.7665
Cope_avoid		0.3255	0.8065
Central sensitisation		0.8138	
Depression		0.7655	
Oxford knee score	0.4221	0.7094	
Patient-specific functional scale			0.3394
Physical activity			

## Data Availability

The original contributions presented in this study are included in the article/Appendix A. Further inquiries can be directed to the corresponding author.

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
