# Peer review of "Preoperative Clinical Phenotyping for Individualised Rehabilitation in End-Stage Knee Osteoarthritis"

_jfmk, 2025, doi:10.3390/jfmk10030360_

Round 1

Reviewer 1 Report

Comments and Suggestions for Authors

In this cross-sectional study, Coetzee and colleagues investigated clinical phenotypes in patients with end-stage knee osteoarthritis (OA) awaiting total knee replacement (TKR) in South Africa. Using exploratory factor analysis on a sample of 72 patients, the authors identify three distinct phenotypic clusters based on modifiable clinical, biomechanical, and psychosocial variables: i) gait and weight associated with obesity, altered gait mechanics, and low self-efficacy; ii) central pain characterised by central sensitisation, depressive symptoms, and poor functional performance; iii) functional – defined by muscular weakness, low physical activity, and functional limitations.

The authors argue that these phenotypes underscore the need for tailored, multidisciplinary preoperative interventions to enhance surgical outcomes and optimize resource allocation in low- and middle-income countries (LMICs).

Title and Abstract

The abstract is informative but slightly verbose and contains minor redundancies (e.g., “such as, including…”). Consider revising some sentences for clarity and conciseness.

Introduction

A clearer distinction between clinical vs molecular phenotyping would enhance conceptual clarity.

Consider discussing the recent findings of Pennestrì et. al (Appl Sci 2024) on predictors of functional outcomes after surgery in a large cohort of hip and knee OA patients.

Cite more recent work on OA phenotypes in LMICs to further justify the setting-specific focus.

Materials and Methods

The reduced sample size (n=72 instead of the planned 85) warrants further discussion regarding potential impact on factor stability.

The rationale for binarizing or trichotomizing continuous variables for EFA (e.g., BMI, GDI) should be briefly reiterated in the main text.

Statistical justification for EFA is incomplete: please report the 

Kaiser–Meyer–Olkin Measure of Sampling Adequacy and results of Bartlett’s test to confirm data suitability for factor analysis.

Results

The “uniqueness” column in Table 2 is not referenced in the narrative; either explain its significance or remove it.

Discussion

The observation that male patients loaded more strongly onto the “central pain” factor contradicts existing evidence and deserves more in-depth interpretation.

Please consider adding remarks on potential generalisability to other healthcare settings or LMIC contexts.

Conclusions

A more apparent separation between descriptive findings and clinical recommendations would help avoid overinterpretation of cross-sectional data.

Supplementary Materials

Supplementary File 2 (STROBE checklist) is mostly complete. However, “n/a” responses for items such as bias and missing data handling should be clarified.

Author Response

22 August 2025

Re: Revisions on manuscript

Dear Editors and reviewers,

Thank you kindly for your response to our submitted manuscript titled Preoperative clinical phenotyping for individualised care in end-stage knee osteoarthritis”. We want to thank the reviewers for taking the time to review our work and providing detailed feedback and corrections to help improve the quality of the manuscript.

We have addressed the comments made by the reviewers and tabulated the response and line numbers below. The changes are highlighted in yellow on the annotated manuscript. Kindly also consider the proposed title change and let us know if this better suit the paper.

Comment

Response

Line of correction

Reviewer 1

The abstract is informative but slightly verbose and contains minor redundancies (e.g., “such as, including…”). Consider revising some sentences for clarity and conciseness.

Thank you, it has been revised to be more concise.

Abstract 12-27

Introduction

A clearer distinction between clinical vs molecular phenotyping would enhance conceptual clarity.

I added two sentences to make it a bit more clear.

46-51

Consider discussing the recent findings of Pennestrì et. al (Appl Sci 2024) on predictors of functional outcomes after surgery in a large cohort of hip and knee OA patients.

Thank you for the recommendation. It is very interesting, but in the context of our study looking at the patients who are waiting for surgery and the factors that may affect the disease/ symptom progression, the predictors of function after surgery in this article appears to have been based on variables that we have not measured. For our future work, this would certainly be a fantastic resource to reference.

Cite more recent work on OA phenotypes in LMICs to further justify the setting-specific focus

Reference added from a 2025 study. Working n paragraph adjusted to justify setting specific focus

77-80

Methods and materials

The reduced sample size (n=72 instead of the planned 85) warrants further discussion regarding potential impact on factor stability.

Thank you. We decided to remove the non-modifiable variables from the factor analysis and ended with 13 variables which required 65 individuals. This improved stability of the factors.

The rationale for binarizing or trichotomizing continuous variables for EFA (e.g., BMI, GDI) should be briefly reiterated in the main text.

Thank you. You will notice that we re-analysed our data with the raw continuous variables (not dichotomised). The authors felt this was more appropriate and it strengthened our results.

Statistical justification for EFA is incomplete: please report the 

Kaiser–Meyer–Olkin Measure of Sampling Adequacy and results of Bartlett’s test to confirm data suitability for factor analysis.

Thank you, it was added.

188-190

Results

The “uniqueness” column in Table 2 is not referenced in the narrative; either explain its significance or remove it.

Removed, thank you

Discussion

The observation that male patients loaded more strongly onto the “central pain” factor contradicts existing evidence and deserves more in-depth interpretation.

This was removed with re-analysis of the variables therefore this is no longer appropriate to comment on as it was most likely not accurate to start with.

Please consider adding remarks on potential generalisability to other healthcare settings or LMIC contexts.

We noted that this was a specialised tertiary hospital which may affect the generalisability of the results to other health care settings. We did add another sentence to comment on some of the results which may be generalisable.

344-349

Conclusion

A more apparent separation between descriptive findings and clinical recommendations would help avoid overinterpretation of cross-sectional data.

Done, thank you

332-341

Supplementary Materials

Supplementary File 2 (STROBE checklist) is mostly complete. However, “n/a” responses for items such as bias and missing data handling should be clarified.

Clarified, thank you

Supp file

Reviewer 2 Report

Comments and Suggestions for Authors

Review:

Preoperative clinical phenotyping for individualised care in end-stage knee osteoarthritis

The manuscript deals with the question about preoperative management for patients with knee osteoarthritis (OA) in South Africa who are awaiting total knee replacement (TKR) surgery. The question is relevant as the number of patients who should receive TKRs is growing and the medical resources are limited.

72 patients were included who were on a list waiting for a TKR surgery and gave consent. Clinical outcomes, functionality, biomechanics and questionnaires were assessed.

Three distinct elements were identified; gait and weight, central pain, functionality.

The authors conclude a heterogeneity in clinical presentations of patients with end-stage OA in South Africa and recommend multidisciplinary preoperative interventions to optimize post-surgical outcomes.

Comments and concerns:

Introduction:

Lines 80-84:

It would be helpful for better understanding to address the aim of the study more exactly; for example starting the sentence with: “The aim of the study was to…..”.

Materials and Methods:

Lines 91 – 105 (“2.2. Study participants”): The explanation of the sample size calculation should be given in the Statistic chapter. Explanations of the situations when the trial was executed (e. g. Covid-19 pandemic leading to a low number of participants) should be discussed in the Discussion chapter.

Line 155: @ VICON: Please complete version, manufacturer, city, country, etc. for the used equipment.

@ Supplementary file1:

p. 198: “….. refined after study 1”. Please explain study 1 and study 2. “*Age categories….”. Please explain the reference of “*” in the figure legend.

Results:

Lines 228 – 243: Factor analysis: Please explain on which considerations the compositions of the 3 different addressed factors: “gait and weight”, “central pain” and “functional” are based. For example: “gait and weight” are presented as major factor of clinical relevance together. It is not clear if the combination of both are addressed or is each of them (gait or weight) is standing on top of the list. What is the input of gait or weight respectively in particular. In addition, this factor (gait and weight) consists of 6 variables (sex, …, self-efficacy…) according table 2 (pp. 6 and 7). Why were the factors bundled? The factor “central pain” consist of 6 variables and interaction between the variables are somewhat arguable. The factor “functional” consists of 3 variable each with understandable input on the functionality. Thus, please clarify the compositions of the 3 addressed factors also from a clinical point of view.  

In this context it is not clear why the variable “Range of motion” is listed under “gait and weight” and “Central pain” in table 2? In my opinion the range of motion should be listed under the factor “functional”. Please clarify.

In the supplementary file1 the stratification shows 12 subgroups regarding gender, age and entities (RA, TOA etc.). It would be helpful to complete the Result chapter with the numbers of assessed patients in terms of allocation to each of these subgroups.

Conclusion:

Lines 349-350: “…..are crucial for optimising outcomes in TKR candidates.”. The outcomes of surgeries in the assessed patient-cohort were not investigated also even the statement is plausible. Thus, the sentence’s place is in the Discussion chapter in my opinion and should be referenced by literature.  

My conclusion: 

The authors present an elaborate study dealing with the status of patients in South Africa suffering from knee OA and waiting for TKR. Except for the addressed limitations (sample size etc.) of the study the compositions of the variables in the 3 factors (gait and weight, central pain and functional) need better explanation and should be reflected also from the clinical point of view and the meaningfulness of the composition.

In addition, a consideration of the input of each subgroup which are given in the figure in the supplementary file 1 to the phenotypes would be helpful; rheumatoid arthritis (RA), traumatic OA and primary OA are probably additional factors which influence the phenotype of patients suffering from OA.

Author Response

22 August 2025

Re: Revisions on manuscript

Dear Editors and reviewers,

Thank you kindly for your response to our submitted manuscript titled Preoperative clinical phenotyping for individualised care in end-stage knee osteoarthritis”. We want to thank the reviewers for taking the time to review our work and providing detailed feedback and corrections to help improve the quality of the manuscript.

We have addressed the comments made by the reviewers and tabulated the response and line numbers below. The changes are highlighted in yellow on the annotated manuscript. Kindly also consider the proposed title change and let us know if this better suit the paper.

Comment

Response

Line of correction

Reviewer 2

Introduction

Lines 80-84:

It would be helpful for better understanding to address the aim of the study more exactly; for example starting the sentence with: “The aim of the study was to…..”.

Thank you, done.

Methods and materials

Lines 91 – 105 (“2.2. Study participants”): The explanation of the sample size calculation should be given in the Statistic chapter. Explanations of the situations when the trial was executed (e. g. Covid-19 pandemic leading to a low number of participants) should be discussed in the Discussion chapter.

This was removed as with re-analysis we removed 3 variables from the factor analysis (age, sex and comorbidities) to only include modifiable variables.

87

Line 155: @ VICON: Please complete version, manufacturer, city, country, etc. for the used equipment.

Added

156

Results

Lines 228 – 243: Factor analysis: Please explain on which considerations the compositions of the 3 different addressed factors: “gait and weight”, “central pain” and “functional” are based. For example: “gait and weight” are presented as major factor of clinical relevance together. It is not clear if the combination of both are addressed or is each of them (gait or weight) is standing on top of the list. What is the input of gait or weight respectively in particular. In addition, this factor (gait and weight) consists of 6 variables (sex, …, self-efficacy…) according table 2 (pp. 6 and 7). Why were the factors bundled? The factor “central pain” consist of 6 variables and interaction between the variables are somewhat arguable. The factor “functional” consists of 3 variable each with understandable input on the functionality. Thus, please clarify the compositions of the 3 addressed factors also from a clinical point of view.  

The composition of the factors (“gait and weight,” “central pain,” and “functional”) was not predetermined but emerged from the exploratory factor analysis (EFA) conducted on the dataset. The naming of each factor reflects the variables with the highest loadings and the underlying clinical construct they were judged to represent. This was added in methods.

200-202

In this context it is not clear why the variable “Range of motion” is listed under “gait and weight” and “Central pain” in table 2? In my opinion the range of motion should be listed under the factor “functional”. Please clarify.

This changed with the removal of the 3 variables as mentioned above. Just to clarify that although clinically it could be argued that ROM belongs under “functional,” the statistical grouping reflected correlations in our dataset. 

In the supplementary file1 the stratification shows 12 subgroups regarding gender, age and entities (RA, TOA etc.). It would be helpful to complete the Result chapter with the numbers of assessed patients in terms of allocation to each of these subgroups.

We agree that this information would be interesting, but are unable to extract that level of information as only the variables inserted into the factor analysis were entered in stata and no link between the diagnosis and the factors can be drawn.

Additional comments

The authors present an elaborate study dealing with the status of patients in South Africa suffering from knee OA and waiting for TKR. Except for the addressed limitations (sample size etc.) of the study the compositions of the variables in the 3 factors (gait and weight, central pain and functional) need better explanation and should be reflected also from the clinical point of view and the meaningfulness of the composition.

As mentioned above, these factors were data driven and not clinically driven. We had no influence over the groupings, only the inclusion of variables, which was decide by the literature on disease and symptom progression.

In addition, a consideration of the input of each subgroup which are given in the figure in the supplementary file 1 to the phenotypes would be helpful; rheumatoid arthritis (RA), traumatic OA and primary OA are probably additional factors which influence the phenotype of patients suffering from OA.

We agree that this information would be interesting, but are unable to extract that level of information as only the variables inserted into the factor analysis were entered in stata and no link between the diagnosis and the factors can be drawn.

Conclusion

Lines 349-350: “…..are crucial for optimising outcomes in TKR candidates.”. The outcomes of surgeries in the assessed patient-cohort were not investigated also even the statement is plausible. Thus, the sentence’s place is in the Discussion chapter in my opinion and should be referenced by literature.  

Agree, thank you. We removed this

Supplementary Materials

p. 198: “….. refined after study 1”. Please explain study 1 and study 2. “*Age categories….”. Please explain the reference of “*” in the figure legend.

Apologies, this was taken from my PhD thesis and I forgot to remove that. I made it clearer.

Supp file

Round 2

Reviewer 1 Report

Comments and Suggestions for Authors

The authors addressed all the issues raised. No further changes are needed.